# Spatial distribution modulation of mixed building blocks in metal-organic frameworks

Seok Jeong[1,2], Junmo Seong[1,2], Sung Wook Moon[1,2], Jaewoong Lim [1], Seung Bin Baek[1], Seung Kyu Min [1✉] & Myoung Soo Lah [1✉]

The placement of mixed building blocks at precise locations in metal–organic frameworks is critical to creating pore environments suitable for advanced applications. Here we show that the spatial distribution of mixed building blocks in metal–organic frameworks can be modulated by exploiting the different temperature sensitivities of the diffusion coefficients and exchange rate constants of the building blocks. By tuning the reaction temperature of the forward linker exchange from one metal–organic framework to another isoreticular metal–organic framework, core–shell microstructural and uniform microstructural metal–organic frameworks are obtained. The strategy can be extended to the fabrication of inverted core–shell microstructures and multi-shell microstructures and applied for the modulation of the spatial distribution of framework metal ions during the post-synthetic metal exchange process of a Zn-based metal–organic framework to an isostructural Ni-based metal–organic framework.

[1] Department of Chemistry, Ulsan National Institute of Science and Technology, Ulsan 44919, Korea. [2] These authors contributed equally: Seok Jeong, Junmo Seong, Sung Wook Moon. ✉email: skmin@unist.ac.kr; mslah@unist.ac.kr

It is important to place mixed building blocks at precise positions in metal-organic frameworks (MOFs) to ensure the suitability of their pore environments for advanced applications[1–3]. It has been reported that multivariate MOFs with uniform spatial distributions of mixed building blocks have properties other than the simple linear sums of the properties of MOFs with pure components[4,5]. The different spatial distributions of mixed building blocks lead to properties that cannot be achieved with mixtures of pure MOFs. Therefore, the spatial distribution modulation of mixed building blocks is important to improve the utility of MOFs.

Post-synthetic exchange (PSE) is an efficient synthetic approach for the preparation of improved MOFs through the replacement of building blocks in MOFs and is reported as a method to control the spatial distribution of building blocks[6–16]. During the process of exchanging building blocks, MOFs of uniform microstructures[17–22] and core–shell microstructures[23–30] were obtained as intermediate species. Matzger et al. reported that the post-synthetic linker exchange of MOF-5 produces core–shell microstructural MOF crystals with different degrees of linker exchange and shell thicknesses depending on the types and molecular sizes of the solvents used[31,32]. In UiO-66, on the other hand, completely opposite results were reported depending on the linker involved and the reaction conditions. While linker exchange in methanol (MeOH) at room temperature produced a uniform microstructural MOF[33], linker exchange in water at 85 °C led to a core–shell microstructural MOF[31]. Meanwhile, Padial and Martí-Gastaldo et al. argued that increasing the linker concentration while keeping the reaction time the same during PSE of UiO-68 resulted in the switching of the spatial linker distribution from core–shell to uniform[34]. The building-block exchange process in MOFs is governed by two processes: diffusion and exchange. Both diffusion and exchange are affected by several factors, such as pore properties, solvents, types and concentrations of the building blocks, and reaction temperatures and times. Therefore, different reaction conditions lead to different spatial distributions of mixed building blocks throughout the MOF crystal. However, the diffusion and exchange kinetics are influenced to different degrees by these factors. The type of reaction solvent and the concentration of entering linker can control the spatial distribution of building blocks to some extent, but factors that can systematically modulate the spatial distribution of building blocks have not been investigated.

For the systematic modulation of the spatial distribution of two mixed linkers, a three-dimensional (3D) MOF, [Ni(HBTC)(AP)] (HAP, where HBTC = benzene-1,3,5-tricarboxylate and AP = azobis(4-pyridine)) based on pillared two-dimensional (2D) sheets[35] is used as a model system (Fig. 1a). Investigation of the linker exchange process using this MOF is advantageous because a reversible pillaring linker exchange from an isoreticular MOF, [Ni(HBTC)(BP)] (where BP = 4,4′-bipyridine), to another had already been reported[36]. The spatial distributions of mixed pillars across the model MOF crystal can be monitored in situ using optical microscopy that exploits the color differences of the mixed pillars and analyzed by Raman mapping of the mixed pillars with different Raman spectra. The kinetics of the forward and reverse pillar exchange processes can be significantly altered by adopting trans-1,2-bis(4-pyridyl)ethene (BE) with a similar linker length but different $pK_a$ value, which can lead to different temperature sensitivities of forward and reverse pillar exchanges (Fig. 1b). The systematic modulation of mixed building blocks can also be applied for the spatial distribution of framework metal ions during the post-synthetic metal exchange process of the 3D MOF, [$Zn_6(BTB)_4(BP)_3$] (ITHD(Zn), where BTB = 4,4′,4″-benzene-1,3,5-tris(benzoate)) to the isostructural 3D MOF, [$Ni_6(BTB)_4(BP)_3$] (ITHD(Ni)). N,N-dimethylformamide (DMF) is chosen as a solvent for all post-synthetic building block

exchanges because both reactants and products are stable in DMF, and DMF has a low melting point and high boiling point, allowing building block exchange over a wide temperature range.

Here, we report the spatial distribution modulation of two mixed building blocks across a MOF crystal via PSE. The modulation is achieved by controlling the reaction temperature in accordance with the different temperature sensitivities of the diffusion coefficients and exchange rate constants of the building blocks in line with the same Arrhenius equation[37]. When the exchange of the building blocks is more sensitive to temperature than the diffusion, then, at higher temperatures, exchange is faster than diffusion, so diffusion-limited pillar exchange produces concentric core–shell microstructured MOF crystals. On the other hand, at lower temperatures, diffusion is faster than exchange, so kinetics-controlled pillar exchange produces uniform microstructured MOF crystals.

## Results

**Reversible pillar exchanges of pillared 3D MOF**. The 3D MOF, HAP, was prepared via a de novo solvothermal reaction with slight modifications to the reported synthetic procedure[35]. This MOF features 3D solvent pores and has two types of portals. The first type of portal with the dimensions of $2.9 \times 3.8$ Å$^2$ lies along the crystallographic [0 0 1] direction, while the other type with the dimensions of $4.7 \times 9.7$ Å$^2$ is located along the crystallographic [1 0 0] and [1 1 0] directions (Fig. 1a). Both the BE and AP pillars can only diffuse into the solvent pores through a portal parallel to the crystallographic ab-plane because the smallest second minimum dimension (MIN-2)[38], ~6.5 Å, of the two pillars is considerably larger than the portal dimension along the crystallographic [0 0 1] direction. Pillar exchange from AP to BE was performed by soaking HAP crystals (~100 mg) in 20 mL 0.3 M BE–DMF solution at 100 °C for 2 days. Although the yellowish-brown crystals turn cyan (Supplementary Fig. 1), the powder X-ray diffraction (PXRD) pattern of [Ni(HBTC)(BE)] (HBE) is highly similar to that of HAP (Supplementary Fig. 2), as the length of the entering BE pillar is approximately the same as that of the leaving AP pillar. The $^1$H nuclear magnetic resonance (NMR) spectrum of HBE digested in a DCl/D$_2$O/dimethyl sulfoxide-d$_6$ (DMSO-d$_6$) solvent mixture confirmed the complete exchange from AP pillars to BE pillars (Supplementary Fig. 3). The reverse pillar exchange from BE to AP was performed by soaking ~100 mg HBE crystals in 20 mL 0.3 M AP–DMF solution at 100 °C for 2 days (Supplementary Fig. 1). The PXRD pattern of HAP obtained via PSE is identical to that of HAP obtained through the de novo solvothermal reaction (Supplementary Fig. 2). The $^1$H NMR spectrum of the HAP crystals obtained through PSE confirmed that the BE pillars were almost completely exchanged by the AP pillars (Supplementary Fig. 3).

**Pillar exchange kinetics and thermodynamics**. The temperature-dependent forward pillar exchange kinetics from HAP to HBE was investigated by soaking ~2 mg of well-ground HAP crystals in 1.5 mL 0.3 M BE–DMF solution (Fig. 2a and Supplementary Fig. 4). The conversion percentage was calculated from the mole fraction of BE in the $^1$H NMR spectrum of the intermediate species obtained after soaking for a given time (Supplementary Figs. 5–8). Almost complete pillar exchange (98% conversion) is observed after 36 and 2 h of soaking at 30 and 50 °C, respectively. Furthermore, 20 min of soaking at 70 °C leads to 100% conversion, and 96% conversion occurs upon soaking for 2 min at 100 °C (Fig. 2a). The reverse pillar exchange from HBE to HAP was also investigated by soaking ~2 mg of HBE crystals in 1.5 mL 0.3 M AP–DMF solution (Fig. 2b and Supplementary Figs. 9–12). The reverse pillar exchange requires significantly more time than the

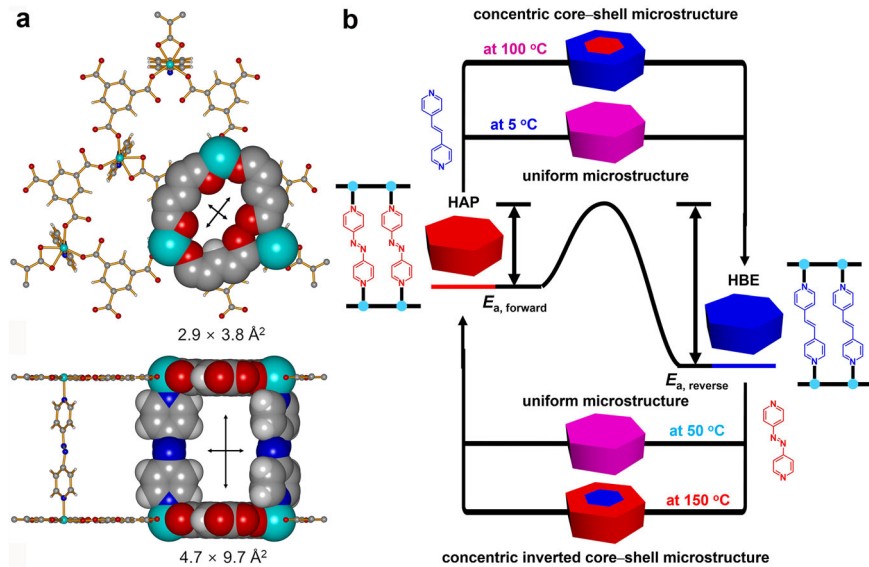

**Fig. 1 Pillared 3D MOF and reversible pillar exchange processes. a** The pillared 3D MOF, [Ni(HBTC)(AP)] (HAP, where HBTC = benezene-1,3,5-tricarboxylate and AP = azobis(4-pyridine)), with **hms** topology illustrating portals along the [0 0 1] (top) and [1 0 0] (bottom) directions. **b** Temperature-controlled reversible pillar exchange between HAP and [Ni(HBTC)(BE)] (HBE, where BE = *trans*-1,2-bis(4-pyridyl)ethene) via concentric core–shell microstructural intermediates and uniform microstructural intermediates. In (**a**), cyan, red, gray, and light gray of the ball-and-stick and space-filling models represent the nickel, oxygen, carbon, and hydrogen atoms, respectively. In the schematic drawings of the concentric core–shell microstructure, concentric inverted core–shell microstructure, and uniform microstructure in (**b**), red and blue indicate the heterogeneous distribution of AP and BE pillars, and pink indicates homogeneous distribution of AP and BE pillars.

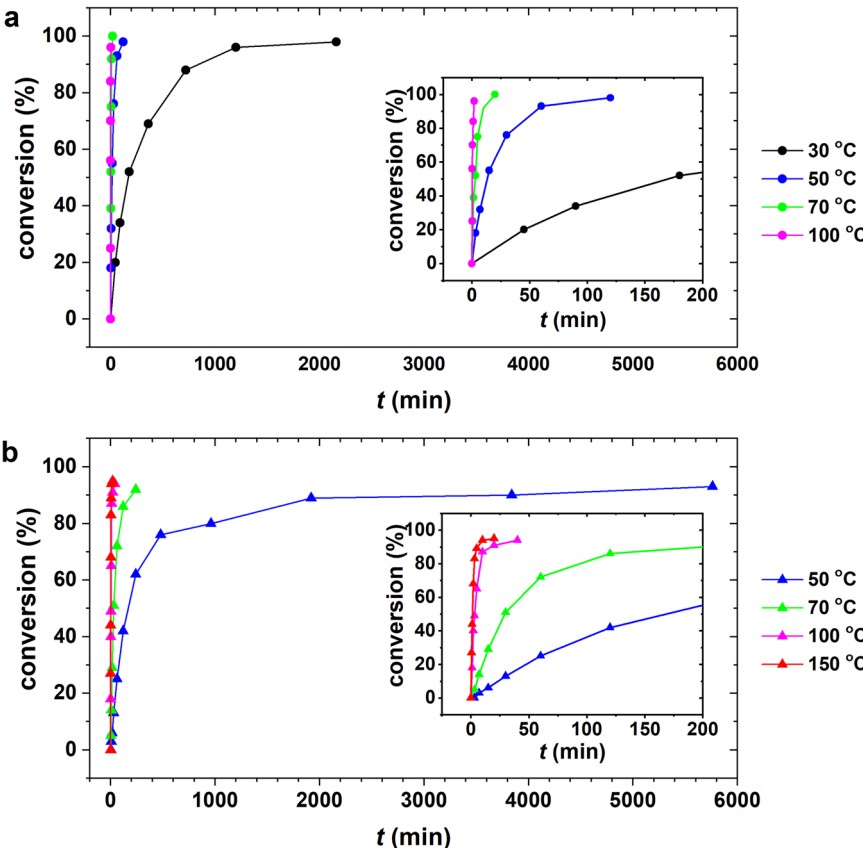

**Fig. 2 Temperature-dependent forward and reverse pillar exchanges. a** Forward pillar exchange from AP to BE at 30, 50, 70, and 100 °C, respectively. **b** Reverse pillar exchange from BE to AP at 50, 70, 100, and 150 °C, respectively.

forward pillar exchange at the same temperature. For example, ~93% pillar conversion is observed after the reaction is allowed to continue at 50 °C for 96 h, which implies that it is more than 50 times slower than the corresponding forward process at the same temperature. At 100 °C, the reverse pillar exchange requires 40 min for 94% conversion, which indicates that it is more than 20 times slower than the corresponding forward pillar exchange process at the same temperature.

The energy difference $\Delta E$ of the forward pillar exchange reaction from HAP to HBE, −12.4(3) kJ/mol, is obtained from the $\ln(K)$ versus $1/T$ plot using temperature-dependent equilibrium constants over the temperature range of 70–150 °C (Supplementary Figs. 13 and 14). This negative $\Delta E$ value is comparable to the energy difference between reactants (HAP crystal with unligated BE) and products (HBE crystal with unligated AP), −11.7 kJ/mol, calculated by density functional theory using HAP and HBE model structures. Therefore, the forward pillar exchange from HAP to HBE is faster than the pillar diffusion due to the relatively low activation energy of the former. The low activation energy of the forward pillar exchange from thermodynamically less stable HAP to thermodynamically more stable HBE is rationalized using the Evans–Polanyi relationship, which states that the activation energy of a reaction is proportional to its enthalpy[39].

**Concentric core–shell and uniform microstructures**. While the degrees of pillar exchange were monitored via the mole fractions of the pillars calculated from the $^1$H NMR spectra of the intermediate MOF, HAP@HBE$_x$($T$,$t$)/HBE@HAP$_x$($T$,$t$), for the core–shell MOF or HAP–HBE$_x$($T$,$t$)/HBE–HAP$_x$($T$,$t$) for the uniform MOF (where $T$ is the temperature in degrees Celsius, $t$ in the format $t$s/$t$m/$t$h/$t$d/$t$w is the soaking time in seconds/minutes/ hours/days/weeks, and $x$, if specified, is the mole fraction of BE/ AP in the intermediate MOF crystals) (Supplementary Fig. 15), the distributions of the pillars across the crystals were monitored by observing the color changes in the crystals using an optical microscope (Supplementary Figs. 1 and 16) and via Raman mapping of the pillars (Supplementary Fig. 16). The forward exchange process was monitored through Raman mapping of the intermediate crystals obtained by soaking ~2 mg of the as-synthesized HAP crystals in 1.5 mL 0.3 M BE–DMF solution (Fig. 3). The optical photographs of the intermediate crystals obtained at 100 °C exhibit concentric core–shell microstructures representing the surface-to-core substitution of the pillars (Supplementary Fig. 17a). The Raman maps confirm the concentric core–shell distribution of the pillars across the crystal (Fig. 3a). A core–shell microstructure is generated because the pillar exchange is much faster than diffusion at 100 °C. In other words, the pillar exchange is a diffusion-limited process. The concentric microstructure, which is characterized by a concentric distribution of pillars, results from the diffusion of the entering pillars (MIN-2: 6.5 Å) into the 3D solvent pores through the portals of dimension $4.7 \times 9.7$ Å$^2$, but not through those of dimension $2.9 \times 3.8$ Å$^2$ (Fig. 1a).

Since the temperature sensitivities of the diffusion and exchange processes of the pillars are different, it is possible to control the distribution of the pillars over the entire crystal by controlling the reaction temperature. Optical photographs and Raman maps of the intermediate crystals obtained at 5 °C show that the two different pillars are uniformly distributed throughout the crystal (Fig. 3b and Supplementary Fig. 17b). The uniform microstructural features indicate that pillar exchange is much slower than pillar diffusion, that is, pillar exchange is a kinetics-dependent process. The uniform microstructural features of the crystal are maintained throughout the pillar exchange process. As the exchange proceeds, the mole fraction of AP decreases in the

crystal while that of BE increases. In concentric core–shell crystals, the concentration gradients of the pillars are further modulated by controlling the reaction temperature. Exchanging the pillars at 50 °C for 30 min leads to a conversion of ~62% from AP to BE (Supplementary Figs. 18 and 19). The extent of this pillar conversion is similar to those observed upon performing the exchange for 3 min at 100 °C (59%) and for 3 days at 5 °C (58%). However, the crystals obtained after 30 min of soaking at 50 °C exhibit a different pillar concentration gradient that lies between those of highly heterogeneous concentric core–shell crystals obtained upon performing the exchange at 100 °C for 3 min and uniform crystals obtained upon soaking for 3 days at 5 °C (Supplementary Fig. 20). Utilizing the different temperature sensitivities of the pillar diffusion and exchange processes, the pillar distribution is modulated across the crystal, leading to the transformation of a highly concentric core–shell distribution to a homogenous uniform distribution.

The reverse pillar exchange from the thermodynamically more stable HBE to the thermodynamically less stable HAP was performed under the same conditions used for the preparation of HBE except for the soaking of the HBE crystals in 1 M AP–DMF solution. The reverse pillar exchange at 100 °C produced a concentric inverted core–shell microstructure (Fig. 4a, Supplementary Figs. 21 and 22). However, the concentration gradient of the pillars was considerably smaller than that observed in the forward exchange reaction performed at the same temperature. Concentric inverted core–shell crystals with a larger pillar concentration gradient are obtained when the reverse pillar exchange is performed at a temperature higher than the forward pillar exchange. Concentric inverted core–shell crystals with a similar concentration gradient are obtained at 150 °C. At this temperature, pillar exchange was considerably faster than pillar diffusion, leading to a highly heterogeneous inverted core–shell microstructural MOF. On the other hand, reverse pillar exchange at 50 °C produced a uniform microstructure, which was similar to that obtained upon performing forward pillar exchange at 5 °C. At 50 °C, the reverse exchange from the thermodynamically more stable HBE to thermodynamically less stable HAP requires more time than the diffusion of the BE pillars into the inner pores.

The strategy for the preparation of concentric core–shell microstructural MOFs (HAP@HBE) utilizing the diffusion-limited pillar exchange was further extended to the production of concentric multi-shell microstructural MOFs (HAP@HBE@-HAP). A concentric multi-shell microstructural MOF, HAP@-HBE@HAP, was prepared by first soaking the HAP crystals in a BE–DMF solution for 4 min at 100 °C, followed by soaking the crystals in an AP–DMF solution for 30 s at 150 °C (Supplementary Fig. 23). Raman mapping of the crystal clearly proved the formation of a concentric multi-shell microstructural MOF (Fig. 4b). The mole fraction of pillars in the multi-shell microstructural MOF crystal was estimated from the mole fraction of AP and BE in the $^1$H NMR spectrum of the bulk crystals (Supplementary Fig. 24). Using the HBE crystal as the starting MOF crystal, a concentric inverted multi-shell microstructural MOF (HBE@HAP@HBE) was prepared in a similar manner (Fig. 4c and Supplementary Fig. 24).

**Kinetic Monte Carlo (kMC) simulation**. The temperature-dependent forward and reverse pillar exchange processes were investigated using a kMC simulation. A hexagonal grid cell was built as a 3D framework model of HAP and HBE. Figure 5 shows a series of snapshots depicting the changes in the spatial distribution of the ligated pillars as a function of normalized time during the forward and reverse pillar exchange processes. For the forward pillar exchange process at 0 °C, there is a uniform increase and decrease in

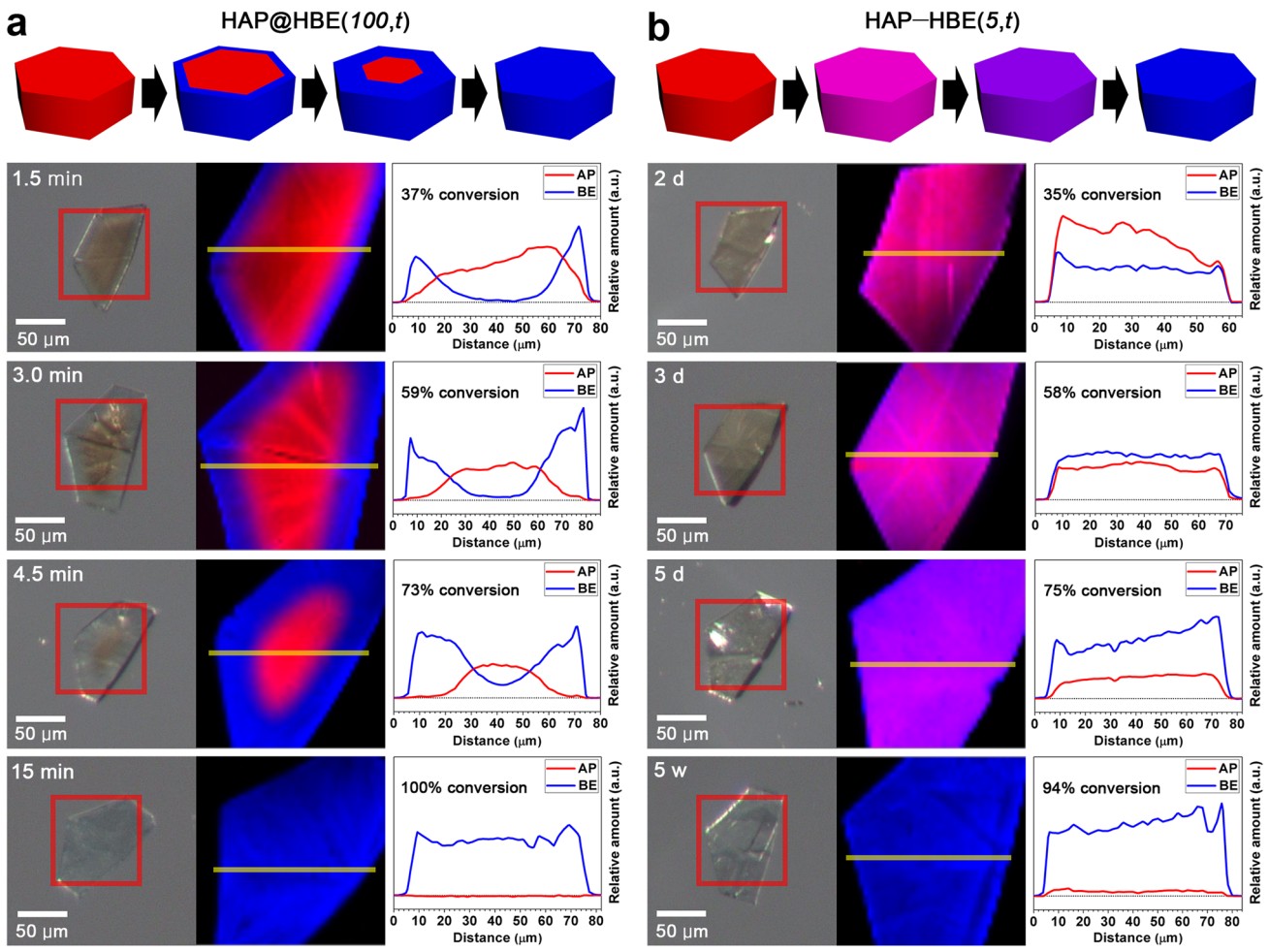

**Fig. 3 Raman maps of AP and BE pillars during forward pillar exchange from HAP to HBE. a** Schematic diagrams, optical photographs, Raman maps, and relative amounts of AP and BE pillars showing concentric pillar distribution in the core–shell intermediates, HAP@HBE$_{0.37}$(100,1.5m), HAP@HBE$_{0.59}$(100,3m), and HAP@HBE$_{0.73}$(100,4.5m), during pillar exchange from HAP to HAP@HBE$_{1.00}$(100,15m) (i.e., HBE). **b** Schematic diagrams, optical photographs, Raman maps, and relative amounts of AP and BE pillars showing uniform pillar distributions with different mole fractions of pillars in the uniform intermediates, HAP–HBE$_{0.35}$(5,2d), HAP–HBE$_{0.58}$(5,3d), and HAP–HBE$_{0.75}$(5,5d), during pillar exchange from HAP to HAP–HBE$_{0.94}$(5,5w). In the schematic diagrams and Raman maps of the crystal, red and blue indicate the distribution of AP and BE pillars, respectively, while pink and purple indicate the homogeneous distribution of AP and BE pillars at different mole ratios throughout the crystal.

the spatial pillar distribution over time (Fig. 5a and Supplementary Fig. 25a). On the other hand, at temperatures exceeding 50 °C, the core–shell pillar distribution is noticeable. To account for the behavior of free leaving pillar after pillar exchange, the effect of the "stay" probability, $\rho$, on the spatial pillar distribution was investigated, where $\rho$ is the probability of a free leaving pillar staying near the site where pillar exchange takes place without spreading toward the reservoir of the hexagonal grid cell. The simulated spatial pillar distributions with three different $\rho$ values of 1, 0.8, and 0 at 50 °C exhibit no significant differences in the spatial distributions of the ligated pillars (Supplementary Fig. 26a). Negligible effect is exerted by $\rho$ on the spatial pillar distribution for the forward pillar exchange process. On the other hand, the spatial pillar distribution in the reverse pillar exchange process at 100 °C is highly dependent on $\rho$ (Supplementary Fig. 26b). A clear core–shell pillar exchange is observed when $\rho = 0$, that is, when the free pillar does not stay near the exchange site after pillar exchange, whereas, when $\rho = 1$, uniform pillar exchange is observed. Simulations for the reverse pillar exchange process with $\rho = 0.8$ predict a more uniform pillar distribution at lower temperatures, and a core–shell pillar distribution with a larger concentration gradient at higher temperatures (Fig. 5b and Supplementary Fig. 25b). The results of the simulation

performed at 100 °C with $\rho = 0.8$ agree with the experimentally observed pillar distribution in the core–shell microstructural crystal.

In both the forward and reverse exchange processes, uniform pillar exchange tends to occur at lower temperatures and core–shell pillar exchange occurs at higher temperatures. The transition temperature from uniform to core–shell pillar exchange in the forward pillar exchange process is lower than that in the reverse pillar exchange process. The core–shell pillar exchange predominates as the temperature increases. The temperature dependence of the pillar exchange behavior originates from the competition between pillar exchange and pillar diffusion (Fig. 5c). While performing simulations, the temperature dependence of the diffusion coefficient was assumed to be significantly smaller than that of the exchange rate constant, and hence, ignored. The overall simulation results are consistent with the experimental observations.

**Control of spatial distribution of framework metal ions**. In addition to the spatial distribution of the organic linkers, the spatial distribution of framework metal ions can be modulated throughout the MOF crystal via PSE. Utilizing the different temperature sensitivities of the exchange and diffusion of metal ions through solvent

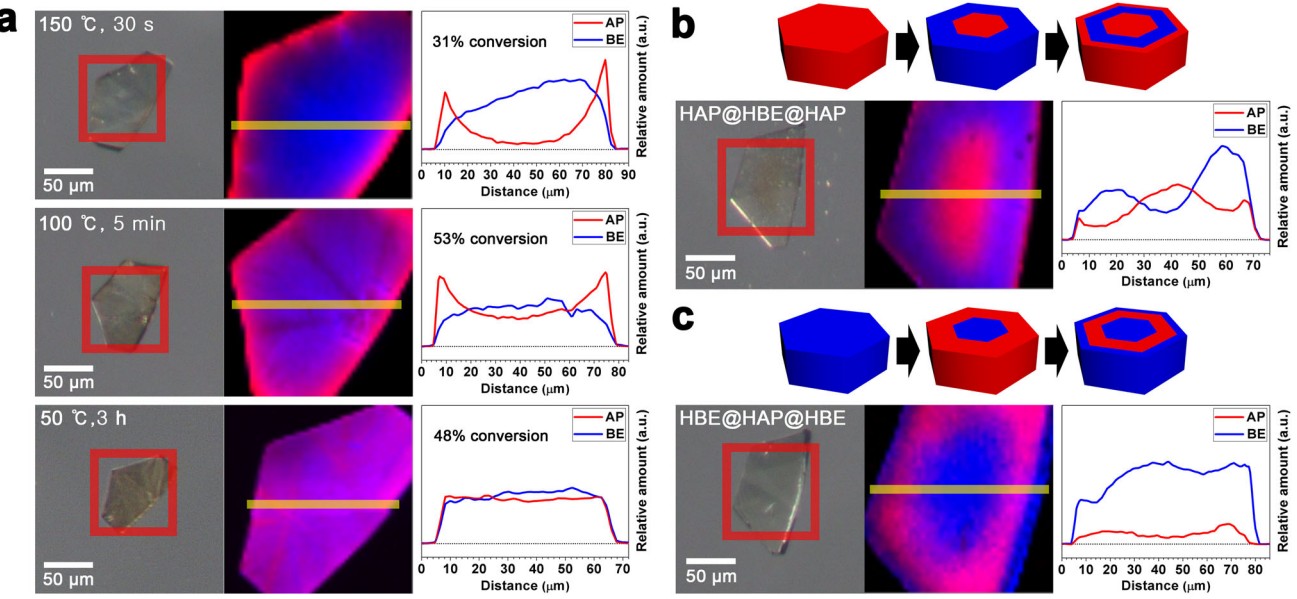

**Fig. 4 Raman maps of AP and BE pillars during reverse pillar exchange from HBE to HAP and fabrication processes of multi-shell microstructures. a** Optical photographs, Raman maps, and relative amounts of AP and BE pillars exhibiting a highly heterogeneous concentric inverted core–shell distribution of the pillars in the intermediate, $HBE@HAP_{0.31}(150,30s)$, a moderately heterogeneous concentric inverted core–shell distribution of the pillars in the intermediate, $HBE@HAP_{0.53}(100,5m)$, and a uniform distribution of the pillars in the intermediate, $HBE–HAP_{0.48}(50,3h)$. **b** Fabrication process of concentric multi-shell microstructural $HAP@HBE(100,4m)@HAP_{0.40}(150,30s)$ and its optical photograph, Raman map, and relative amounts of AP and BE pillars across the crystals. **c** Fabrication process of concentric inverted multi-shell microstructural $HBE@HAP(150,1m)@HBE_{0.87}(100,30s)$, and its optical photograph, Raman map, and relative amounts of AP and BE pillars across the crystals. In the schematic diagrams and Raman maps of the crystal, red and blue indicate the distribution of AP and BE pillars, respectively, while pink and purple indicate the homogeneous distribution of AP and BE pillars at different mole ratios.

pores, it is possible to obtain not only core–shell but also uniform microstructural MOF crystals. As reported, ITHD(Zn) containing Zn(II) ions as framework metal ions can be converted to iso-structural ITHD(Ni) containing Ni(II) ions as framework metal ions by soaking ITHD(Zn) crystals in 0.1 M $Ni(NO_3)_2·6H_2O$–DMF solution at 100 °C for 2 days[40] (Fig. 6a, b). Soaking the crystals under the same conditions for 20 min results in core–shell microstructural MOF crystals because framework metal ion exchange is a diffusion-limited process at 100 °C. The progress of the framework metal ion exchange was monitored via analysis of the crystals for their metal composition using inductively coupled plasma-optical emission spectrometry (ICP-OES). According to the results obtained, soaking the crystals for 20 min under the same conditions produced mixed-metal MOF crystals containing Ni at a mole fraction of 0.30. Optical photographs of the mixed-metal MOF single crystal and its frag-ments clearly showed the core–shell microstructural features of the MOF crystal (Fig. 6c). Upon examination of a cross-sectioned single crystal using scanning electron microscopy (SEM) and energy dis-persive spectroscopy (EDS), further evidence was found for the formation of the core–shell microstructural MOF crystal, ITHD(Zn) $@ITHD(Ni)_{0.30}(100,20m)$ (where 100, 20m, and 0.30 are the soaking temperature in degrees Celsius, soaking time in minutes, and mole fraction of Ni, respectively) (Fig. 6e). On the other hand, instead of soaking the ITHD(Zn) crystals at 100 °C for 20 min, soaking them in the same Ni-DMF solution at 5 °C for 4 days resulted in mixed-metal MOF crystals containing Ni at a mole fraction of 0.17. However, the spatial distribution of framework metal ions in this mixed-metal MOF crystal is completely different from that in the crystal obtained by soaking at 100 °C for 20 min. Optical photo-graphs of a single crystal and its fragments clearly show the uniform microstructural features of the MOF crystal (Fig. 6d). The results of SEM imaging and EDS mapping of a cross-sectioned single crystal

also support the formation of the uniform microstructural MOF crystal, $ITHD(Zn)–ITHD(Ni)_{0.17}(5,4d)$ (Fig. 6f).

## Discussion

We demonstrated a general strategy for the spatial distribution modulation of mixed building blocks, mixed organic linkers and mixed framework metal ions, utilizing the different temperature sensitivities of the diffusion and exchange of the building blocks. The forward pillar exchange process from HAP to HBE crystals at a high temperature produced core–shell microstructural MOF crystals with a large concentration gradient of pillars across the crystal as inter-mediate species. The forward pillar exchange from HAP to HBE is considerably faster than the pillar diffusion due to the relatively low activation energy of the forward pillar exchange. On the other hand, the same forward pillar exchange process when conducted at a low temperature produced uniform microstructural MOF crystals with uniform pillar concentrations across the crystal. The concentration gradient of the pillars is tuned by adjusting the reaction temperature. Similarly, on account of the higher activation energy of the reverse pillar exchange process, MOF crystals of inverted core–shell microstructures are obtained via reverse pillar exchange from HBE to HAP at a temperature higher than that of the forward pillar exchange process. Uniform MOF crystals can also be obtained through a reverse exchange at a low temperature exceeding that of the forward pillar exchange process. The strategy for the preparation of concentric core–shell microstructural MOFs can be further extended to the production of concentric multi-shell microstructural MOFs, HAP@HBE@HAP and HBE@HAP@HBE, which have alternating shell regions comprising different pillars. This strategy can also be applied for the spatial distribution modulation of fra-mework metal ions through PSE utilizing the different temperature sensitivities of the diffusion and exchange of metal ions. This

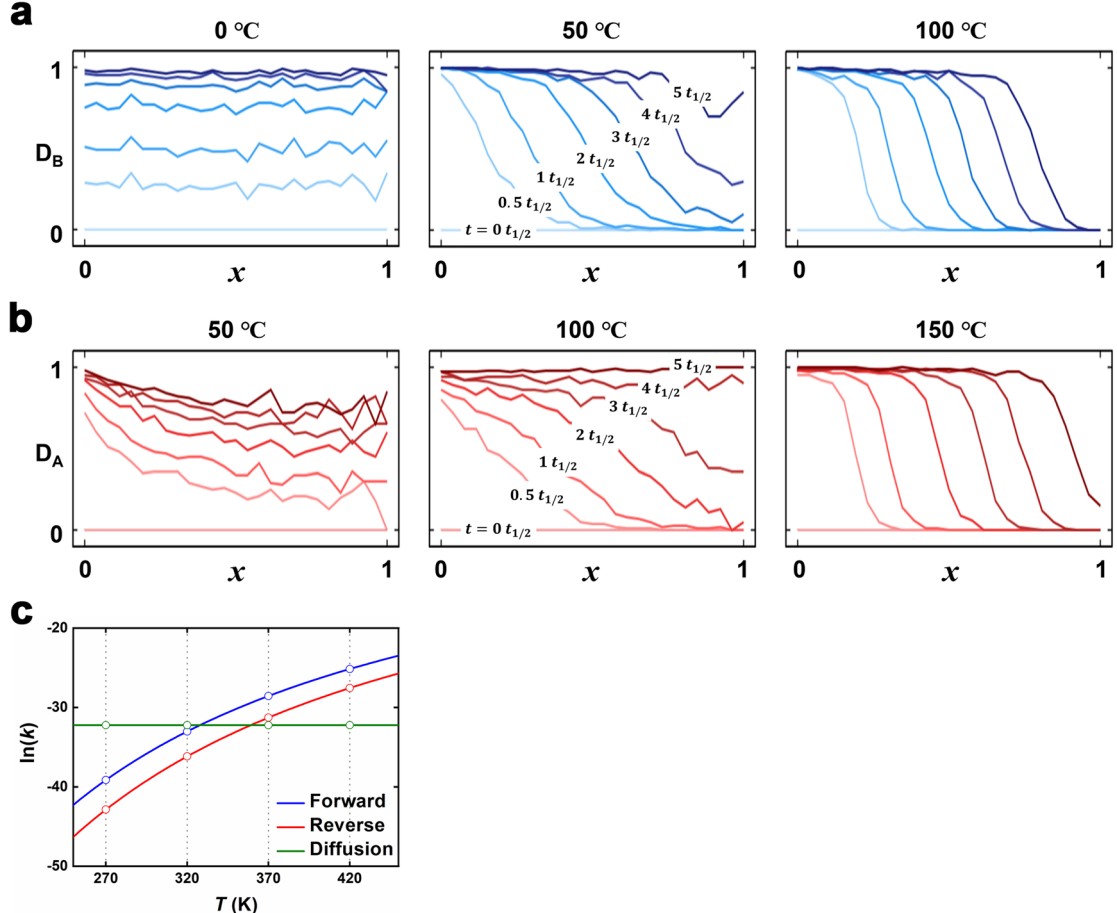

**Fig. 5 Mean spatial distributions, $D_{MAP/MBE}(x,t;T)$, of AP ligated to a metal (MAP) and BE ligated to a metal (MBE) in the hexagonal cell. a** $D_{MAP/MBE}(x,t;T)$ during forward pillar exchange at 0, 50, and 100 °C over radial factor $x$ and normalized time $t$. **b** $D_{MAP/MBE}(x,t;T)$ during reverse pillar exchange at 50, 100, and 150 °C over radial factor $x$ and normalized time $t$. Red and blue lines in **a** and **b** represent the mean spatial distributions of MAP and MBE, respectively. **c** Temperature dependence of the rate constant ($k$) for kMC simulations in the range of 0–150 °C.

efficient approach to achieve the spatial distribution modulation of mixed building blocks provides a straightforward synthetic tool for the development of MOF materials suitable for use in a variety of applications, such as catalysts, sensors, and molecular separations.

## Methods
**General procedures**. All reagents were purchased from commercial sources and used without further purification. PXRD data ($2\theta$ angle) were recorded using a Bruker D2 Phaser automated diffractometer at room temperature with a step size of 0.02°. $^1H$ NMR spectra were obtained using an Agilent 400-MR DD2 NMR spectrometer operating at 400 MHz for $^1H$. Raman mapping was performed with a WITec alpha300 R Confocal Raman microscope equipped with a 532 nm laser and a 600 g/mm grating. Data analysis was conducted using WITec Project 2.10. Cold field-emission SEM (Cold FE-SEM) was performed using a Hitachi SU-8220 at an accelerating voltage of 15 kV. EDS elemental mapping images were captured using an Octane Elect EDS system (Ametek EDAX) to confirm the distribution of metallic elements. MOF crystals were epoxy-fixed at 70 °C overnight and cut with an EM UC7 ultramicrotome (Leica) prior to the SEM and EDS analyses.

**Preparation of MOFs via de novo solvothermal reaction**
*Preparation of [Ni(HBTC)(AP)]·4DMF·H₂O (HAP·4DMF·H₂O).* HAP crystals with a dimension of ~50 μm were prepared via a reported procedure[35] with slight modifications. Ni(NO₃)₂·6H₂O (0.032 g, 0.11 mmol), 1,3,5-benzene tricarboxylic acid (H₃BTC, 0.021 g, 0.10 mmol), and AP (0.028 g, 0.15 mmol) were dissolved in a mixture of DMF/MeOH (19 mL, 10:9 ratio). The solution was heated in a tightly sealed 30 mL vial at 60 °C for 15 h to form yellowish green hexagonal plate-shaped crystals with dimensions of ~50 μm. The crystals were washed 5 times over 2 days, each time with 10 mL fresh DMF to remove any residual reactants and linkers present in the solvent pores. The crystals were harvested and dried for 1 h under ambient conditions (Yield, 0.013 g, 17%).

*Preparation of [Zn₆(BTB)₄(BP)₃]·46DMF·35H₂O, (ITHD(Zn)·46DMF·35H₂O).* ITHD(Zn) crystals were prepared according to a reported procedure[40]. Zn(NO₃)₂·6H₂O (0.0382 g, 0.128 mmol), H₃BTB (0.0442 g, 0.101 mmol), and BP (0.0086 g, 0.055 mmol) were dissolved in 5 mL DMF in an 8 mL glass vial. The solution was then heated in an isotherm oven at 70 °C for 1 day to form colorless rhombic dodecahedral crystals, which were collected via filtration and washed with fresh DMF. The crystals were harvested and dried for 1 h under ambient conditions (yield, 0.031 g, 26%).

**Preparation of MOFs via post-synthetic linker exchange**. The crystals synthesized in bulk via the PSE procedure were washed at least five times over 2 days with 10 mL fresh DMF each time, and the crystals harvested for Raman mapping via the PSE procedure were washed at least four times (each time with 4 mL fresh DMF) over 1 day after each soaking step.

*Preparation of [Ni(HBTC)(BE)] (HBE).* Approximately 100 mg of HAP crystals were added to 20 mL 0.3 M BE–DMF solution in a 30 mL vial. The vial was tightly sealed and stored in a 100 °C oven for 2 days. The solution containing the HAP crystals was refreshed once with 20 mL fresh 0.3 M BE–DMF solution during soaking.

*Preparation of HAP.* Approximately 100 mg of HBE crystals were added to 20 mL 0.3 M AP–DMF solution in a 30 mL vial. The vial was tightly sealed and stored in a 100 °C oven for 2 days. The solution containing the HBE crystals was refreshed once with 20 mL fresh 0.3 M AP–DMF solution during soaking.

**Preparation of concentric core–shell MOFs**
*Concentric core–shell HAP@HBEₓ(100,t).* Approximately 5 mg of ~50 μm-sized HAP crystals were soaked in 1.5 mL 0.3 M BE–DMF solution at 100 °C for a given time $t$. The solution was immediately refreshed using 4 mL fresh DMF and dried for 2 h under ambient conditions. The ratio of the pillars in the concentric core–shell HAP@HBEₓ(100,t) crystals (where $x$ represents the mole fraction of BE pillars) was modulated by controlling the soaking time. The mole fraction of BE

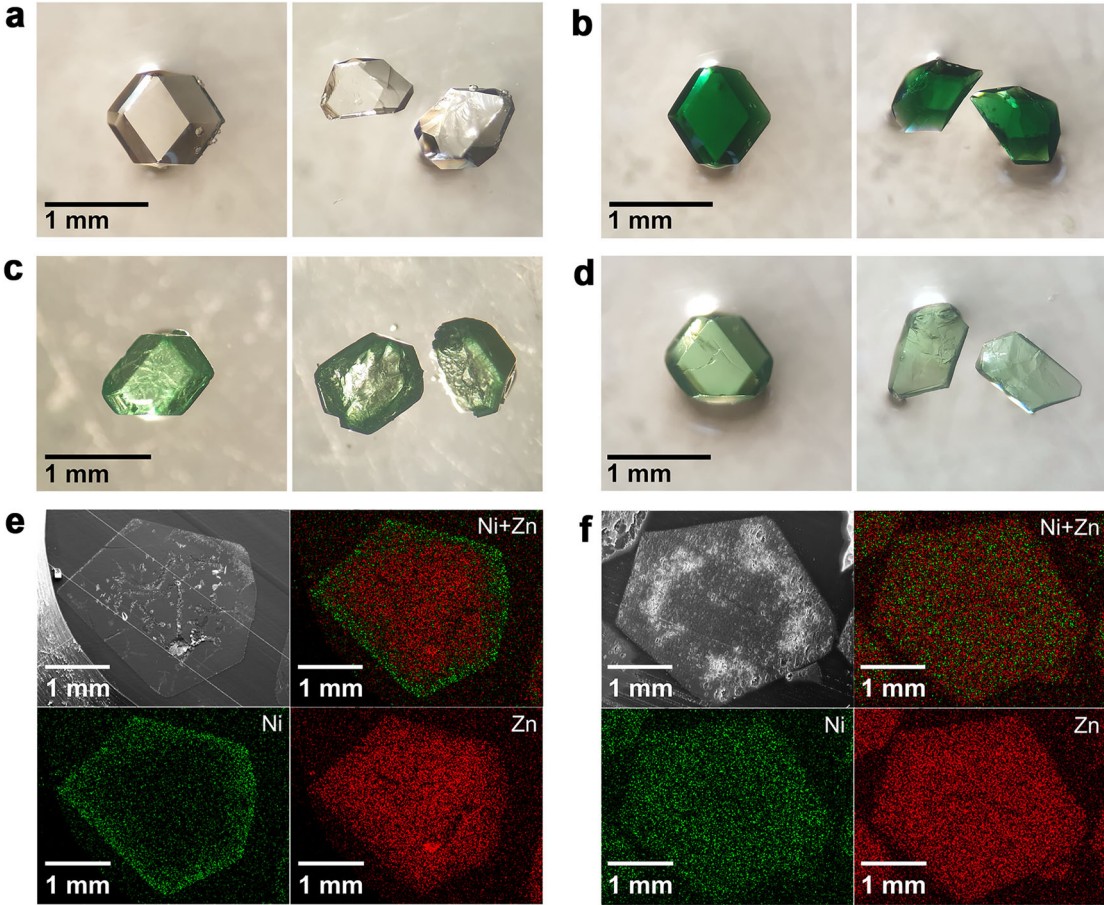

**Fig. 6 Optical photographs, SEM images and EDS maps of core–shell ITHD(Zn)@ITHD(Ni) and uniform ITHD(Zn)–ITHD(Ni) crystals. a** Optical photographs of as-synthesized ITHD(Zn) single crystal and its fragments. **b** Optical photographs of ITHD(Ni) single crystal obtained via PSE and its fragments. **c** Optical photographs of ITHD(Zn)@ITHD(Ni)$_{0.30}$(100,20m) core–shell crystal and its fragments. **d** Optical photographs of ITHD(Zn)–ITHD(Ni)$_{0.17}$(5,4d) uniform crystal and its fragments. **e** SEM image and EDS maps of cross-sectioned ITHD(Zn)@ITHD(Ni)$_{0.30}$(100,20m) crystal showing core–shell distribution of metal ions. **f** SEM image and EDS maps of cross-sectioned ITHD(Zn)–ITHD(Ni)$_{0.17}$(5,4d) crystal showing uniform distribution of metal ions.

pillars in the concentric core–shell crystals was obtained from the $^1$H NMR spectrum of the crystals digested in 0.65 mL DCl/D$_2$O/DMSO-d$_6$ mixed solvent (0.05 mL 1.0 M aqueous DCl, 0.4 mL D$_2$O, and 0.2 mL DMSO-d$_6$).

*Concentric inverted core–shell HBE@HAP$_x$(150,t).* Approximately 5 mg of ~50 μm-sized HBE crystals were soaked in 1.5 mL 1.0 M AP–DMF solution at 150 °C for a given time *t*. The solution was immediately refreshed using 4 mL fresh DMF and dried for 2 h under ambient conditions. The ratio of the pillars in the concentric inverted core–shell HBE@HAP$_x$(150,t) crystals (where *x* represents the mole fraction of AP pillars) was modulated by controlling the soaking time. The mole fraction of BE pillars in the concentric inverted core–shell crystals was obtained from the $^1$H NMR spectrum of the crystals digested in 0.65 mL DCl/D$_2$O/DMSO-d$_6$ mixed solvent (0.05 mL 1.0 M aqueous DCl, 0.4 mL D$_2$O, and 0.2 mL DMSO-d$_6$).

### Preparation of uniform MOFs
*Uniform HAP–HBE$_x$(5,t).* Approximately 5 mg of ~50 μm-sized HAP crystals were soaked in 1.5 mL 0.3 M BE–DMF solution at 5 °C for a given time *t*. The solution was immediately refreshed using 4 mL fresh DMF and dried for 2 h under ambient conditions. The ratio of the pillars in the uniform HAP–HBE$_x$(5,t) crystals (where *x* represents the mole fraction of BE pillars) was modulated by controlling the soaking time. The mole fraction of BE pillars in the uniform crystals was obtained from the $^1$H NMR spectrum of the crystals digested in 0.65 mL DCl/D$_2$O/DMSO-d$_6$ mixed solvent (0.05 mL 1.0 M aqueous DCl, 0.4 mL D$_2$O, and 0.2 mL DMSO-d$_6$).

*Uniform HBE–HAP$_x$(50,t).* Approximately 5 mg of ~50 μm-sized HBE crystals were soaked in 1.5 mL 1.0 M AP–DMF solution at 50 °C for a given time *t*. The solution was immediately refreshed using 4 mL fresh DMF and dried for 2 h under ambient conditions. The ratio of the pillars in the uniform HBE–HAP$_x$(50,t) crystals (where *x* represents the mole fraction of AP pillars) was modulated by controlling the soaking time. The mole fraction of AP pillars in the uniform crystals was obtained from the $^1$H

NMR spectrum of the crystals digested in 0.65 mL DCl/D$_2$O/DMSO-d$_6$ mixed solvent (0.05 mL 1.0 M aqueous DCl, 0.4 mL D$_2$O, and 0.2 mL DMSO-d$_6$).

### Preparation of concentric multi-shell MOFs
*Concentric multi-shell HAP@HBE(100,4m)@HAP$_{0.40}$(150,30s).* For the preparation of concentric multi-shell crystals, ~5 mg of HAP crystals were first soaked in 1.5 mL 0.3 M BE–DMF solution at 100 °C for 4 min. The solution was immediately refreshed using 4 mL DMF to remove any unligated pillars remaining in the solvent pores. The crystals were then transferred to 1.0 M AP–DMF solution at 150 °C and aged for 30 s.

*Concentric inverted multi-shell HBE@HAP(150,1m)@HBE$_{0.87}$(100,30s).* For the preparation of concentric inverted multi-shell crystals, ~5 mg of HBE crystals were first soaked in 1.5 mL 1.0 M AP–DMF solution at 150 °C for 1 min. The solution was immediately refreshed using 4 mL DMF to remove any unligated pillars remaining in the solvent pores. The crystals were then transferred to 0.3 M BE–DMF solution at 100 °C and aged for 30 s.

### Preparation of MOFs via post-synthetic metal exchange
*Preparation of [Ni$_6$(BTB)$_4$(BP)$_3$] (ITHD(Ni)).* Approximately 30 mg of colorless ITHD(Zn) crystals, refreshed using DMF, were soaked in 1.0 mL 0.1 M Ni(NO$_3$)$_2$·6H$_2$O DMF solution at 100 °C for 2 days.

*Preparation of core–shell ITHD(Zn)@ITHD(Ni)$_{0.30}$(100,20m).* Approximately 30 mg of ITHD(Zn) crystals were soaked in 1.0 mL 0.1 M Ni(NO$_3$)$_2$·6H$_2$O-DMF solution at 100 °C for 20 min. The solution was immediately refreshed using 4 mL fresh DMF, and the harvested crystals were dried under ambient conditions for 2 h. The mole fraction of Ni in the core–shell MOF crystals was determined using the ICP-OES analysis results of Zn and Ni.

*Preparation of uniform ITHD(Zn)–ITHD(Ni)$_{0.17}$(5,4d)*. ~30 mg of ITHD(Zn) crystals were soaked in 1.0 mL 0.1 M Ni(NO$_3$)$_2$·6H$_2$O-DMF solution at 5 °C for 4 days. The solution was immediately refreshed using 4 mL fresh DMF, and the harvested crystals were dried under ambient conditions for 2 h. The mole fraction of Ni in the uniform MOF crystals was determined using the ICP-OES analysis results of Zn and Ni.

## Pillar exchange kinetics

*Forward pillar exchange reaction*. The HAP crystals were wet-ground in the presence of DMF to reduce the diffusion effect due to the large crystal size (Supplementary Fig. 4a) and dried in an oven at 60 °C for ~1 min. ~2 mg of the dried HAP crystals were transferred to a 4 mL vial containing 1.5 mL 0.3 M BE–DMF solution preheated to a given temperature. After reaction for a given time, the crystals were thoroughly washed four times over 1 day using fresh DMF each time. The progress of the pillar exchange reaction was monitored from the mole fractions of the pillars calculated from the $^1$H NMR spectra of the intermediate crystals.

*Reverse pillar exchange reaction*. The HBE crystals were wet-ground in the presence of DMF to reduce the diffusion effect due to the large crystal size (Supplementary Fig. 4b) and dried in an oven at 60 °C for ~1 min. ~2 mg of the dried HBE crystals were transferred to a 4 mL vial containing 1.5 mL 0.3 M AP–DMF solution preheated to a given temperature. After reaction for a given time, the crystals were thoroughly washed four times over 1 day using fresh DMF each time. The progress of the pillar exchange reaction was monitored from the mole fractions of the pillars calculated from the $^1$H NMR spectra of the intermediate crystals.

## Pillar exchange thermodynamics

A 4.0 mg amount of wet-ground HAP crystals dried at 60 °C for 1 min was prepared in a 4 mL vial. After the addition of a 0.90 mL $1.0 \times 10$ mM BE–DMF solution pre-heated at 150 °C into the vial, the vial was placed at 150 °C oven for a given time. After the sedimentation of the crystalline powder using centrifugation, the crystalline powder was quickly washed using fresh 1.5 mL DMF twice. After the additional soaking the crystals in fresh 1.5 mL DMF for 12 h, the samples were collected and dried at 60 °C for 5 min. The dried sample was digested in 0.05 mL D$_2$SO$_4$ and then 0.6 mL D$_2$O/DMSO-d$_6$ mixed solution (0.4 mL/0.2 mL) was added. The mole fractions of AP and BE in the crystalline powder were analyzed using a $^1$H NMR spectrum of the digested sample. The same experiment was repeated at 120, 100 and 70 °C, respectively.

## Raman mapping

Mixed-pillar intermediate MOF crystals for Raman mapping were obtained by soaking the HAP or HBE crystals under specified conditions. The crystals were extensively washed using fresh DMF to remove unligated BE and AP pillars from the solvent pores and dried under ambient conditions for 2 h. The mole fractions of the pillars in the mixed-pillar crystals used for Raman mapping studies were estimated from the $^1$H NMR spectra of the crystals digested in DCl/D$_2$O/DMSO-d$_6$ mixed solvent. The peak at $1150-1180$ cm$^{-1}$ in the Raman spectrum of the HAP crystals was assigned to the AP pillars, and the peak at $1630-1650$ cm$^{-1}$ was assigned to the BE pillars. In the Raman map, they are decoded as red and blue, respectively.

## KMC simulation

KMC simulations of pillar exchange reactions were performed using a home-made program based on MonteCoffee[41], modified as follows. A 2D triangular cell with a dual-grid scheme is built as a 3D framework model of HAP/HBE to mimic the realistic molecular situations of pillar diffusion and exchange. The triangular cell was further expanded into a hexagonal cell system by applying a periodic boundary condition, where the two edges of a triangular cell face the symmetry-related edges of the adjacent triangular cells, and the other remaining edge serves as the diffusion boundary of the free pillar. In the dual-grid scheme, each exchange grid (EG) point can have two different exchange states, MAP and MBE, where MAP and MBE represent the state of an AP ligated to a metal ion (M–AP) and that of a BE ligated to a metal ion (M–BE), respectively. Meanwhile, each diffusion grid (DG) point can have three different diffusion states (0, AP, and BE): the absence of both AP and BE (i.e., the vacancy of the sites intended for occupation by pillars or the presence of solvent alone), the presence of AP, and the presence of BE. An exchange reaction only occurs if two different pillars are located at the same grid point. To account for the behavior of free leaving pillar after pillar exchange through the forward and reverse pillar exchange processes, the effect of the "stay" probability, $\rho$, was introduced, where $\rho$ is defined as the probability of a free leaving pillar remaining at a DG point without spreading toward the nearest vacant neighboring DG points on the reservoir side of the hexagonal grid cell. The temperature-dependent kinetic constant for exchange reaction was adopted based on the Arrhenius equation with pre-exponential factor ($A$). The energy barriers for forward and reverse exchanges were set at 21 and 23 kcal/mol, respectively, to account for the experimental results. The diffusion rate constant of the free pillar from one DG point to the nearest neighboring DG point at the diffusion grid was set to $10^{-14}$ A.

## Data availability

Data to support the findings of this study can be found in the manuscript, supplementary information, or from the authors upon request.

## Code availability

MonteCoffee code modified for kMC simulations for this study is available upon request from the corresponding authors.

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

## Acknowledgements

This work was supported by grants (2016R1A5A1009405) from the National Research Foundation (NRF) of Korea.

## Author contributions

S.J., J.S., and M.S.L. conceived the idea and designed the experiments. S.J., J.S., J.L., and S.B.B. synthesized and characterized the materials. S.W.M. and S.K.M. performed kMC simulations for the PSE process of HAP and HBE. All authors discussed the results, analyzed the data, and commented on the manuscript. M.S.L. supervised the overall study.

## Competing interests

The authors declare no competing interests.
