## [Peer Review File · Nature Communications]

Spatial Distribution Modulation of Mixed Building Blocks in Metal–Organic FrameworksREVIEWER COMMENTS

Reviewer #1 (Remarks to the Author):

This is a very clear and readable paper that brings core-shell functionalization studies up a notch in rigor. The main contribution is to carefully probe the role of temperature. My expectation would have been rates would change but not microstructure. Wrong! They demonstrate the ability to change microstructure and do so with lots of beautiful quantitative data and strong mechanistic discussion. The metal exchange work is also very nice. The kinetic data is a valuable contribution (Figure 1 is a treasure trove) and the modeling adds significantly as well. The work complements prior work where solvent leads to a change from core-shell to uniform functionalization albeit with less dramatic results than obtained here.

It is also notable that the core-shell functionalization is conducted in a non-cubic MOF so the functionalization is anisotropic. This is uncommon or, perhaps, previously unknown. The authors may wish to comment on this point.

The only fault I can find in the study is that really only one structure type was studied and it is not one with much inherent importance. This can be overlooked considering the quality of the study.

Reviewer #2 (Remarks to the Author):

The manuscript authored by Min and Lah described a general strategy to control the spatial distribution of building blocks in MOFs. Through elaborate kinetics control, a few core-shell MOF structures can be obtained. The researchers also confirmed that this strategy could be applied in both ligand exchange and metal exchange. Design and preparation of hierarchical MOF-on-MOF structures have been a long-term challenge. This work will provide a practical approach to engineering predesigned MOF structures. Herein, I recommend that the manuscript should be accepted with minor revision.

1. The authors systematically study the kinetics of the ligand exchange process, which involves a competition of ligand exchange and diffusion. Their conclusions are supported by solid evidence, such as Raman mapping and computation simulation. Could the authors design experiments to calculate the energy difference between the two MOFs to depict a more detailed energy diagram?
2. The authors designed a general and powerful approach to construct hierarchical core-shell MOFs. Some significant works in this field should be cited. For example, Liu and coauthors observed a similar phenomenon in a 2D MOF system. (*J. Am. Chem. Soc.* 2020, 142, 4705–4713.) Feng and coworkers gave systematic reviews on hierarchical MOFs. (*Chem. Soc. Rev.*, 2019, 48, 4823-4853; *ACS Cent. Sci.* 2018, 4, 1719–1726.)
3. Can this strategy be utilized in the linker exchange of longer organic linkers?
4. In Figure 5, the authors utilized the EDS mapping to study the distribution of Zn and Ni. But the EDS can only tell the distribution of elements on the surface. Therefore, it is recommended that the authors should cut the crystal to perform the EDS again.

Reviewer #3 (Remarks to the Author):

MOFs are fascinating materials with various functional properties that depend on building blocks. Mixing of building blocks is one of the strategy to improve functional properties of MOFs. However, special distribution of mixed building blocks can be very different depending on the synthesis method, solvent and temperature. Therefore, it is important to find a way how to place the mixed building blocks at precise positions in MOFs. The authors report very detailed study of a few mixed-block MOFs, both core-shell and uniform, and discuss processes that control the building blocks and metal cations exchange. The results are interesting, sound, original and of interest for very broad scientific audience. I think, however, that the presentation of the results could be improved:

1. Introduction: the authors wrote that “The factors that can modulate the spatial distribution of building blocks have not been investigated.” Literature data report synthesis of both uniform and core-shell MOFs and in fact some information on factors affecting distribution of building blocks is given in literature. I think that the authors should mention about this and emphasize that in the present report, they study these effects in a systematic way.
2. Introduction: the third paragraph is too long and sounds in many places like conclusions. This paragraph should contain more general information. For instance, what factors can be used to control the spatial distribution of the building blocks and which of them the authors used and why? why DMF or DMF/MeOH were used instead of other solvents? which experimental methods can be employed to obtain information on the spatial distribution and why the authors decided to use optical and Raman methods?
3. Results and Discussion section is difficult to follow. I recommend to add subsections, maybe experimental data for HAP/HBE system (normal and reverse exchange of the building blocks), experimental multi-shell, Monte Carlo simulations, control of spatial distribution of framework metal ions.
4. Different temperatures were used to obtain various structures. It is clear that low temperatures lead to uniform MOFs and high to core-shell MOFs. However, why in some cases 70 or 60 degree C and in other 100 degree C or 150 degree C were used? Are these temperatures optimal or they were chosen based on literature reports?
5. There are some typos in the experimental section (degrees are seen as squares)

Responses to Reviewer Comments

Reviewer #1 (Remarks to the Author):

This is a very clear and readable paper that brings core-shell functionalization studies up a notch in rigor. The main contribution is to carefully probe the role of temperature. My expectation would have been rates would change but not microstructure. Wrong! They demonstrate the ability to change microstructure and do so with lots of beautiful quantitative data and strong mechanistic discussion. The metal exchange work is also very nice. The kinetic data is a valuable contribution (Figure 1 is a treasure trove) and the modelling adds significantly as well. The work complements prior work where solvent leads to a change from core-shell to uniform functionalization albeit with less dramatic results than obtained here.

It is also notable that the core-shell functionalization is conducted in a non-cubic MOF so the functionalization is anisotropic. This is uncommon or, perhaps, previously unknown. The authors may wish to comment on this point.

Response: Thanks for the suggestion. However, core-shell functionalization in non-cubic MOFs has already been reported. Reference # 24 described the post-synthetic incorporation of Ni on the external surface of non-cubic CPO-27(Mg).

The only fault I can find in the study is that really only one structure type was studied and it is not one with much inherent importance. This can be overlooked considering the quality of the study.

Response: We are currently working on other pillared 3D MOFs. Preliminary results also indicate that the pillar distributions can be modulated by controlling the reaction temperature.

Reviewer #2 (Remarks to the Author):

The manuscript authored by Min and Lah described a general strategy to control the spatial distribution of building blocks in MOFs. Through elaborate kinetics control, a few core-shell MOF structures can be obtained. The researchers also confirmed that this strategy could be

applied in both ligand exchange and metal exchange. Design and preparation of hierarchical MOF-on-MOF structures have been a long-term challenge. This work will provide a practical approach to engineering pre-designed MOF structures. Herein, I recommend that the manuscript should be accepted with minor revision.

1. The authors systematically study the kinetics of the ligand exchange process, which involves a competition of ligand exchange and diffusion. Their conclusions are supported by solid evidence, such as Raman mapping and computation simulation. Could the authors design experiments to calculate the energy difference between the two MOFs to depict a more detailed energy diagram?

Response: The energy difference ΔE between HAP and HBE, $-12.4(3)$ kJ/mol, is calculated from the slope ($-\Delta E/R$) of the $\ln(K)$ versus $1/T$ plot obtained using temperature-dependent equilibrium constants of HAP to HBE pillar exchange reaction over the temperature range of 70–150 °C (Supplementary Figs. S13 and S14). This negative ΔE value is comparable to the energy difference between reactants (HAP crystal with unligated BE) and products (HBE crystal with unligated AP), -11.7 kJ/mol, calculated by density functional theory using HAP and HBE model structures (See Supplementary Information for details).

2. The authors designed a general and powerful approach to construct hierarchical core-shell MOFs. Some significant works in this field should be cited. For example, Liu and coauthors observed a similar phenomenon in a 2D MOF system. (J. Am. Chem. Soc. 2020, 142, 4705–4713.) Feng and coworkers gave systematic reviews on hierarchical MOFs. (Chem. Soc. Rev., 2019, 48, 4823–4853; ACS Cent. Sci. 2018, 4, 1719–1726.)

Response: According to the reviewer's suggestion, we have included the above-mentioned papers as references.

3. Can this strategy be utilized in the linker exchange of longer organic linkers?

Response: Although we have not attempted to use longer organic linkers, we believe that this strategy can be utilized for linker exchange of longer organic linkers if the exchange kinetics is fast enough within the temperature ranges of investigation.

4. In Figure 5, the authors utilized the EDS mapping to study the distribution of Zn and Ni. But the EDS can only tell the distribution of elements on the surface. Therefore, it is recommended that the authors should cut the crystal to perform the EDS again.

Response: Figs. 5e and 5f show the EDS mapping of cross-sections of crystals cut using an ultramicrotome. As described in the last sentence of the general procedure in the Methods section, MOF crystals were epoxy-fixed at 70 °C overnight and cut with an ultramicrotome prior to the SEM and EDS analyses.

Reviewer #3 (Remarks to the Author):

MOFs are fascinating materials with various functional properties that depend on building blocks. Mixing of building blocks is one of the strategy to improve functional properties of MOFs. However, special distribution of mixed building blocks can be very different depending on the synthesis method, solvent and temperature. Therefore, it is important to find a way how to place the mixed building blocks at precise positions in MOFs. The authors report very detailed study of a few mixed-block MOFs, both core-shell and uniform, and discuss processes that control the building blocks and metal cations exchange. The results are interesting, sound, original and of interest for very broad scientific audience. I think, however, that the presentation of the results could be improved:

1. Introduction: the authors wrote that “The factors that can modulate the spatial distribution of building blocks have not been investigated.” Literature data report synthesis of both uniform and core-shell MOFs and in fact some information on factors affecting distribution of building blocks is given in literature. I think that the authors should mention about this and emphasize that in the present report, they study these effects in a systematic way.

Response: It was noted that the type of reaction solvent and the concentration of entering linker can control the spatial distribution of building blocks to some extent, but there has been no systematic investigation of the modulation the spatial distribution of building blocks. The relevant description has been revised according to the reviewer's suggestion.

2. Introduction: the third paragraph is too long and sounds in many places like conclusions. This paragraph should contain more general information.

Response: Following the reviewer's suggestion, we removed explanations such as conclusion-like descriptions and included experimental methods to control and analyze the spatial distribution of building blocks and solvent selection.

For instance, what factors can be used to control the spatial distribution of the building blocks and which of them the authors used and why?

Response: As described in the second sentence of the third paragraph, the spatial distribution of the building blocks across a MOF crystal can be modulated by controlling the reaction temperature in accordance with the different temperature sensitivities of the diffusion coefficients and exchange rate constants of the building blocks following the same Arrhenius equation. Experimental results show that the exchange rate constant of the building blocks is more sensitive to temperature than the diffusion coefficient. Therefore, at higher temperatures, exchange is faster than diffusion, so diffusion-limited pillar exchange produces concentric core-shell microstructured MOF crystals. On the other hand, at lower temperatures, diffusion is faster than exchange, so kinetics-controlled pillar exchange produces uniform microstructured MOF crystals.

Why DMF or DMF/MeOH were used instead of other solvents?

Response: HAP was prepared in a DMF/MeOH mixed solvent. On the other hand, all post-synthetic pillar exchanges in this study were performed under DMF. DMF was chosen because both HAP and HBE are stable in DMF, and DMF has a melting point of $-61\text{ }^{\circ}\text{C}$ and a boiling point of $153\text{ }^{\circ}\text{C}$, allowing pillar exchange over a relatively wide temperature range.

Which experimental methods can be employed to obtain information on the spatial distribution and why the authors decided to use optical and Raman methods?

Response: The spatial distribution of the ligated pillars is somewhat ambiguous due to the unligated pillars remaining in the solvent pores, but since the color of HAP crystals is different from that of HBE crystals, the progress of pillar exchange can be monitored in situ using optical microscopy. Since the Raman spectrum of the entering pillar (BE) is different from that of the leaving pillar (AP), the spatial distribution of the ligated mixed pillars throughout the MOF crystal can be easily mapped.

3. Results and Discussion section is difficult to follow. I recommend to add subsections, maybe experimental data for HAP/HBE system (normal and reverse exchange of the building blocks), experimental multi-shell, Monte Carlo simulations, control of spatial distribution of framework metal ions.

Response: Thanks for the suggestion. According to your suggestion, several subsection headings such as reversible pillar exchanges of pillared 3D MOF, pillar exchange kinetics and thermodynamics, concentric core-shell and uniform microstructures, kinetic Monte Carlo simulation and control of spatial distribution of framework metal ions were added to the Results and Discussion section.

4. Different temperatures were used to obtain various structures. It is clear that low temperatures lead to uniform MOFs and high to core-shell MOFs. However, why in some cases 70 or 60 degree C and in other 100 degree C or 150 degree C were used? Are these temperatures optimal or they were chosen based on literature reports?

Response: As explained in Discussion section, the forward pillar exchange at a high temperature produced core-shell microstructural MOF crystals with a large concentration gradient of pillars across the crystal as intermediate species as the forward pillar exchange is considerably faster than the pillar diffusion due to the relatively low activation energy of the forward pillar exchange. On the other hand, the same forward pillar exchange process when conducted at a low temperature produced uniform microstructural MOF crystals with uniform pillar concentrations across the crystal as the pillar exchange is much slower than the pillar diffusion. The optimal temperatures for concentric core-shell microstructures and homogeneous uniform microstructures were chosen from experimental trials.

5. There are some typos in the experimental section (degrees are seen as squares)

Response: All typographical errors have been fixed.

REVIEWER COMMENTS

Reviewer #2 (Remarks to the Author):

The authors have addressed all of my comments in the revision. The manuscript can be published in its current form.

Reviewer #3 (Remarks to the Author):

The authors addressed all issues raised by the reviewers and I recommend publication of this paper in Nature Communications.

REVIEWERS' COMMENTS

Reviewer #2 (Remarks to the Author):

The authors have addressed all of my comments in the revision. The manuscript can be published in its current form.

Response: There are no further comments for the revision.

Reviewer #3 (Remarks to the Author):

The authors addressed all issues raised by the reviewers and I recommend publication of this paper in Nature Communications.

Response: There are no further comments for the revision.